# IvRA: A Framework to Enhance Attention-Based Explanations for Language Models with Interpretability-Driven Training

**Sean Xie**
Department of Computer Science
Dartmouth College
sean.xie.gr@dartmouth.edu

**Soroush Vosoughi**[*]
Department of Computer Science
Dartmouth College
soroush.vosoughi@dartmouth.edu

**Saeed Hassanpour**[*]
Department of Biomedical Data Science
Dartmouth College
saeed.hassanpour@dartmouth.edu

## Abstract

Attention has long served as a foundational technique for generating explanations. With the recent developments made in Explainable AI (XAI), the multi-faceted nature of interpretability has become more apparent. Can attention, as an explanation method, be adapted to meet the diverse needs that our expanded understanding of interpretability demands? In this work, we aim to address this question by introducing IvRA, a framework designed to directly train a language model's attention distribution through regularization to produce attribution explanations that align with interpretability criteria such as simulatability, faithfulness, and consistency. Our extensive experiments demonstrate that IvRA outperforms existing methods in guiding language models to generate explanations that are simulatable, faithful, and consistenti. In addition, we perform ablation studies to verify the robustness of IvRA across various experimental settings and to shed light on the interactions between different interpretability criteria.

## 1 Introduction

The rapid adoption of language models (Devlin et al., 2018; Liu et al., 2019; Lewis et al., 2019; Achiam et al., 2023) in recent years has sparked an escalating interest in enhancing model interpretability. This has given rise to the burgeoning field of Explainable AI (XAI), which has devised various methods to increase model interpretability (Shrikumar et al., 2016; Ribeiro et al., 2016; Shrikumar et al., 2017). However, an universal definition for the term "interpretability" remains elusive in the research community (Lipton, 2016). Interpretability assessment has primarily leaned on criteria tailored for different purposes that fall under the broad umbrella of the term "Interpretability". Some of the most popular criteria are *simulatability* (Doshi-Velez and Kim, 2017), *faithfulness*

---

[*]Co-corresponding Authors.

(Jacovi and Goldberg, 2020; Ribeiro et al., 2016), and *consistency* (Serrano and Smith, 2019; Jain and Wallace, 2019). Simulatability measures whether a model's behavior is comprehensible enough for a human or another ML model to predict its outputs on unseen data, aligning with the objective of conveying the model's underlying mechanics to humans. Faithfulness measures the extent to which an explanation reflects the actual decision-making process of the model. Consistency assesses the explanation method's stability across varying input data, favoring explanations that remain similar for similar inputs and reflect input changes that lead to inconsistent outputs.

The utility of attention for generating saliency explanations is widely recognized (Deng et al., 2017; Wiegreffe and Pinter, 2019; Vashishth et al., 2019; Martins et al., 2020), notwithstanding initial doubts regarding the faithfulness and consistency of attention mechanisms (Serrano and Smith, 2019; Jain and Wallace, 2019). Past works (Atanasova et al., 2020; Sun et al., 2024) that have benchmarked existing attention-based text attribution methods along interpretability criteria such as simulatability, faithfulness and consistency do not explore the possibility of directly training attention distributions to become more interpretable with regard to a criterion. On the other hand, works that *do* train their explanations to become more interpretable via some criterion either only focus on a small subset of criteria (Pruthi et al., 2022; Neely et al., 2021; Fernandes et al., 2022) and/or do not use attention as a technique (Chan et al., 2022b), instead relying on a separate model as rationale extractor. In this work, we focus on developing an attention-based explanation framework that enables a language model (LM) to produce explanations that align more closely with interpretability criteria. We summarize our contribution below:

This paper introduces a novel framework—**I**nterpretability **v**ia **R**egularized **A**ttention

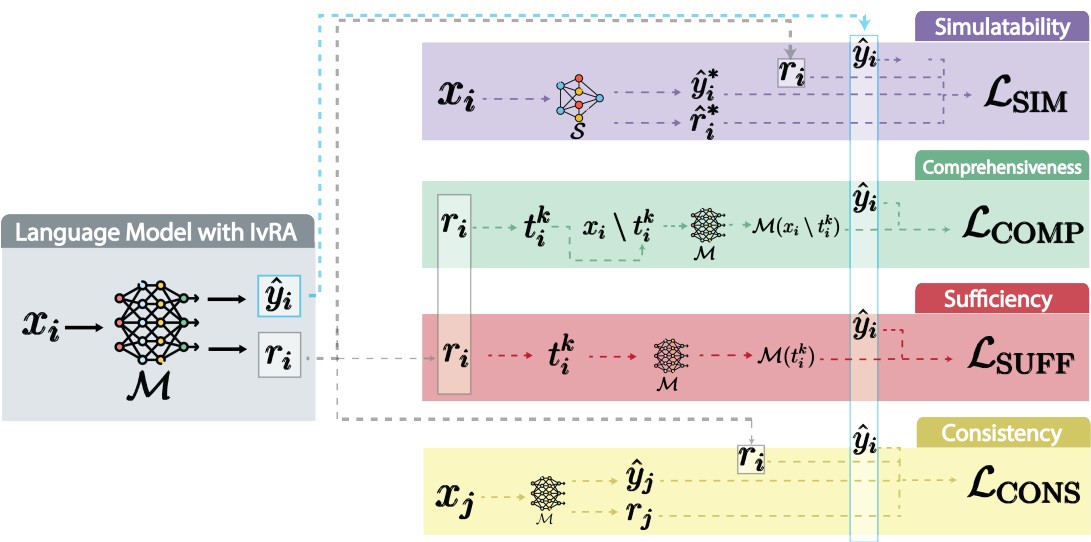

Figure 1: Illustration of IvRA, our proposed framework. A IvRA model ($\mathcal{M}$) takes as input $x_i$ and produces output logits $\hat{y}_i$ along with saliency explanations $r_i$. Different loss functions ($\mathcal{L}$) corresponding to different criteria then take $r_i$ and $\hat{y}_i$ as input to propagate loss back to the model. For simulatability (§2.2.1), $\mathcal{S}$ denotes a student model, with $\hat{y}_i^*$ and $\hat{r}_i^*$ denoting the output logit and explanations of $\mathcal{S}$, respectively. For comprehensiveness and sufficiency (§2.2.2), $t_i^k$ denotes tokens recognized by $r_i$ with attention scores in the top $k\%$ of tokens. For consistency (§2.2.3), $x_j$ represents another example within the dataset.

(IvRA) that parameterizes attention distributions in LMs to produce attention-based attribution explanations ($r_i$) alongside their outputs. The operation of our framework is illustrated in Fig. 1. During training, IvRA uses specialized loss functions for each criterion to propagate losses to a set of weights within the interpretable attention modules of the LM (Fig. 2) in order to optimize the attention-based explanations for each criterion. We empirically verify IvRA's effectiveness in terms of simulatability, faithfulness (comprehensiveness and sufficiency), and consistency[1] on three NLP tasks: Text Classification, Entailment Inference, and Question-Answering. Our results demonstrate that IvRA effectively enhances model interpretability, guiding LMs to generate simulatable, faithful, and consistent explanations for their decisions.

## 2   Background and Methodology [2]

Since our work seeks to integrate various criteria of interpretability for training, the amount of related literature needed to detail our methodology for *each* criterion is extensive. To conserve space, we included only key works that we think are crucial to understanding our contribution. See §A for additional related works.

---

[1]We include a discussion as well as a study on the criterion of *plausibility* using human annotated rationales in §F
[2]We share our source code at github.com/yx131/IvRA-Interpretability-Driven-Training

### 2.1   Interpretable Attention Module

Given an input sequence $x_i$ of length $L$, an attention head $h$ processes $x_i$ through linear projections to yield $Q_i^h$ and $K_i^h$, thereby computing a normalized distribution $\text{Att}_i^h \in \Delta_{L-1}^L = \text{softmax}\{Q_i^h(K_i^h)^T\}$ (Vaswani et al., 2017). Recent research has highlighted the effectiveness of attention-based interpretation methods in enhancing the interpretability of language models (Treviso and Martins, 2020; Kobayashi et al., 2020). Furthermore, because the attention mechanism is instrinsic to the LM, attention-based explanations possess the advantage of not requiring a separate procedure that is decoupled from the decision-making process, in contrast to post-hoc methods (Shrikumar et al., 2016; Du et al., 2019). Building on this foundation, our work seeks to cultivate more interpretable attention-based explanations by parameterizing the multi-head attention layers within a LM and optimizing the parameterized attention distribution in accordance with specified interpretability objectives. In more detail, for the query and key projections $Q_i^h$ and $K_i^h$ of head $h$, we compute normalized feature-wise distributions as shown in equations 1 and 2:

$$\tilde{Q}_i^h = \text{NORM}(\omega_Q^h \odot Q_i^h) \qquad (1)$$

$$\tilde{K}_i^h = \text{NORM}(\omega_K^h \odot K_i^h) \qquad (2)$$

For each layer $\ell$, we compute the distribution $\Psi_\ell$

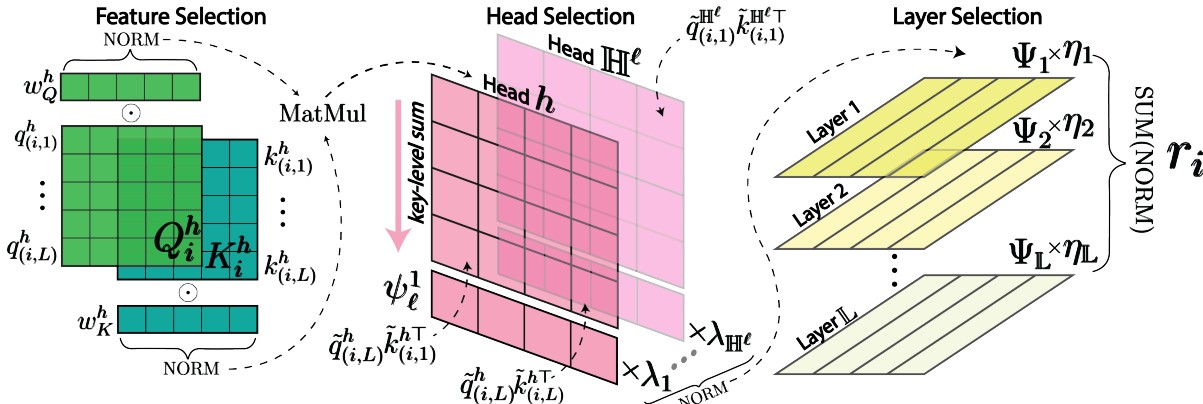

Figure 2: Illustrated architecture of IvRA's interpretable attention module. The end output $r_i$ for each input is a vector of saliency scores, each corresponding to a token in the input $x_i$.

over the attention heads using equation 3, where $\tilde{q}_{(i,n)}^h$ represents the portion of normalized query projection in $\tilde{Q}_i^h$ corresponding to token $n$ in $x_i$ and $\lambda_h$ is a trainable coefficient for each head.

$$\Psi_\ell = \mathsf{NORM}\left(\left[\lambda_h \psi_\ell^h\right]_{h=1}^{\mathbb{H}^\ell}\right) \tag{3}$$

where

$$\psi_\ell^h = \frac{1}{L} \cdot \sum_n^L \tilde{q}_{(i,n)}^h (\tilde{K}_i^h)^T \tag{4}$$

Lastly, to determine the aggregated attention distribution $r_i$, we sum the normalized distribution of all $\Psi_\ell$'s, as defined in equation 5, where $\eta_\ell$ is a coefficient for each layer:

$$r_i = \mathsf{SUM}\left(\mathsf{NORM}\left(\left[\eta_\ell \Psi_\ell\right]_{\ell=1}^{\mathbb{L}}\right)\right) \in \Delta_{L-1} \tag{5}$$

The design of our interpretable attention module, as outlined above, serves dual purposes:

**1) Aggregation for Salience**: In order to derive $r_i \in \Delta_{L-1}$, it's necessary to aggregate the attention distributions across layers and heads. This is because the multi-head attention distribution is a matrix of dimension $L \times L$. Common interpretability measures such as faithfulness and consistency are only applicable to 1-dimensional saliency scores. In the absence of IvRA, it's common to either use the attention heads in the final layer or the mean attention distribution across all layers in the model for layer aggregation (Fomicheva et al., 2020). **2) Optimization for Interpretability**: Our attention module facilitates systematic aggregation through learnable parameters and allows for hyperparameter experimentation, such as the normalizing function for NORM.

Our approach to regularizing attention is similar to the attention-based explainer used in Fernandes et al. (2022) to elicit explanations for a student-teacher setup (SMaT). However, the SMaT explainer is relatively **coarse**, as it only learns weights for head selection. Fernandes et al. (2022) did not explore the effectiveness of feature and layer selection and confined their interpretability evaluation to just *one* criterion (simulatability). In contrast, our framework not only seeks to employ an attention-based explainer that integrates **four** criteria, but also employs the parameterization of attention at the feature, head, and layer levels. Our detailed ablation study in §C, demonstrates that parameterization at all levels is the most effective strategy.

## 2.2 Interpretability Objectives

We formulate our interpretability objectives as distinct loss functions: simulatability, faithfulness (comprehensiveness and sufficiency), and consistency. Given a classification task with $C$ classes, we denote a dataset as $\mathcal{D} = \{(x_i, y_i)\}_{i=1}^N$ consisting of $N$ samples, with each $x_i$ as an input sequence of length $L$ and $y_i$ representing the ground truth label. We denote the output logits of model $\mathcal{M}$ for input $x_i$ as $\mathcal{M}(x_i) \in \mathbb{R}^C$, and the predicted class as $\tilde{y}_i = \mathrm{argmax}(\hat{y}_i)$.

### 2.2.1 Simulatability

Simulatability refers to the capacity of a model to generate decisions replicable by a human observer (Doshi-Velez and Kim, 2017; Lipton, 2016). This interpretability measure proves beneficial by quantifying the efficacy of model behavior communication (Treviso and Martins, 2020). Simulatability is evaluated both through manual annotations (Hase and Bansal, 2020) and automated methods (Pruthi

et al., 2022). In this work, we adopt the automated approach outlined by Pruthi et al. (2022) and extended by Fernandes et al. (2022). Here, simulatability is gauged as the extent to which a *student* model can replicate the *teacher* model's predictions given a saliency explanation of the teacher's input. We employ this simulatability evaluation construct to enhance the simulatability of our primary (i.e., teacher) model $\mathcal{M}$. To this end, we train $\mathcal{M}$ to generate an explanation $r_i$, which we use in the training of a student model $\mathcal{S}$ to replicate $\tilde{y}_i$. Let the output logits $\hat{y}_i^* = \mathcal{S}(x_i)$ of $\mathcal{S}$ for $x_i$, and let $r_i^*$ be the attention module output of $\mathcal{S}$ for $y_i^*$, we define simulatability accuracy on a dataset $\mathcal{D}$ as shown in equation 6.

$$
\mathsf{SIM}(M, S, \mathcal{D}) = \\
\frac{1}{|\mathcal{D}|} \sum_{(x_i, y_i) \in \mathcal{D}} \mathbb{1}\{\tilde{y}_i = \mathrm{argmax}(\hat{y}_i^*)\} \quad (6)
$$

Considering $\tilde{y}_{i,c}$ and $\hat{y}_{i,c}^*$ as values of $\tilde{y}_i$ and $\hat{y}_i^*$ for class $c \in C$, respectively, we define simulatability loss for *a single instance* as the sum of cross-entropy loss between $M$'s predictions and $S$'s predictions and the Kullback–Leibler divergence loss between $\mathcal{M}'s$ and $\mathcal{S}$'s attention outputs (eq. 7).

$$
\mathcal{L}_{\mathsf{SIM}} = \\
\sum_{c \in C} \tilde{y}_{i,c} \log(\hat{y}_{i,c}^*) + \mathrm{KLDiv}(r_i, r_i^*) \quad (7)
$$

It's crucial to note that simulatability should be evaluated under a *constrained* setting, wherein the student's learning capability is intentionally limited. Two frequently employed strategies are: 1) simplifying the student model architecture, or 2) utilizing a distinct data subset for simulatability evaluation, different from that used to train the teacher (Fernandes et al., 2022). We adopt the second strategy in our experiments. For additional information on simulatability, please refer to §B.1 in the appendix.

### 2.2.2 Faithfulness

Faithfulness represents the extent to which an explanation accurately captures the underlying reasoning process of model $\mathcal{M}$ in predicting $\tilde{y}_i$ (Jacovi and Goldberg, 2020). To gauge the faithfulness of our explanations for $\mathcal{M}$, we examine the impact of salient tokens identified by our extracted explanation $r_i$ on $\hat{y}_i$ using comprehensiveness and sufficiency (DeYoung et al., 2019). We define

$t_i$ as the sequence of tokens obtained by binarizing $r_i$ over a $k\%$ threshold, i.e., $t_i^k \in \{0,1\}^L = \left\{ \begin{array}{l} 1 \text{ if } r_i^l \text{ is in in the top } k\% \text{ of salient scores} \\ 0 \text{ else} \end{array} \right\}_{l=1}^L$
Given $p_{\tilde{y}_i}(x_i)$ as $\mathcal{M}$'s confidence probability for $\tilde{y}i$ with input $x_i$, we compute comprehensiveness as the difference in $p_{\tilde{y}_i}$ with $t_i^k$ removed from the input (eq. 8). In essence, if tokens identified by $t_i^k$ are *comprehensive*, their exclusion from the input should decrease the predicted probability of $\mathcal{M}$ for $\tilde{y}_i$. Similarly, we determine sufficiency by calculating the difference in $p_{\tilde{y}_i}$ when only retaining the identified tokens in $t_i^k$ (eq. 9). In this case, tokens in $t_i^k$ are deemed *sufficient* if keeping them as the sole input elements does not reduce $\mathcal{M}$'s predicted probability for $\tilde{y}_i$.

$$
\mathsf{COMP} = p_{\tilde{y}_i}(x_i) - p_{\tilde{y}_i}(x_i \setminus t_i^k) \quad (8)
$$

$$
\mathsf{SUFF} = p_{\tilde{y}_i}(x_i) - p_{\tilde{y}_i}(t_i^k) \quad (9)
$$

In our experiments, we compute COMP and SUFF for each individual $k \in \{1, 5, 10, 20, 50\}$. We calculate the final COMP and SUFF values as the area-over-precision curve (AOPC) for all $k$ values in the set (DeYoung et al., 2019; Chan et al., 2022b). Furthermore, we define comprehensiveness loss for a single instance $x_i$ as the difference between cross-entropy losses when using $x_i$ as input versus $x_i \setminus t_i^k$ as input. This is lower-bounded by a margin $\mu_{\mathsf{comp}}$ to prevent exceedingly high negative losses (eq. 10). Likewise, we define sufficiency loss for a single instance as the difference between cross-entropy losses when using $t_i^k$ as input and $x_i$ as input (eq. 11), lower-bounded by $\mu_{\mathsf{suff}}$. For additional details on faithfulness loss, see B.3.

$$
\mathcal{L}_{\mathsf{COMP}} = \mu_{\mathsf{comp}} + \max\Big\{ -\mu_{\mathsf{comp}}, \\
-\Big( \tilde{y}_i \log(\mathcal{M}(x_i)) - \tilde{y}_i \log(\mathcal{M}(x_i \setminus t_i^k)) \Big) \Big\} \quad (10)
$$

$$
\mathcal{L}_{\mathsf{SUFF}} = \mu_{\mathsf{suff}} + \max\Big\{ -\mu_{\mathsf{suff}}, \\
-\Big( \tilde{y}_i \log(\mathcal{M}(t_i^k)) - \tilde{y}_i \log(\mathcal{M}(x_i)) \Big) \Big\} \quad (11)
$$

### 2.2.3 Consistency

Consistency refers to the ability of explanation methods to produce similar reasoning paths for similar instances of data (Robnik-Šikonja and Bohanec, 2018; Serrano and Smith, 2019; Jain and

Wallace, 2019). Consequently, if two instances $x_i$ and $x_j$ are perceived as similar by $\mathcal{M}$, then $r_i$ and $r_j$, the salient scores provided by IvRA, should also exhibit similarity. We note that our focus is on the similarity of interpretations in $r_i$ and $r_j$, not on the similarity of outcomes. Identical predictions do not necessarily imply analogous model reasoning, which is the essence of our interest in consistency. We derive $\mathcal{H}_i$, the aggregate hidden state for $x_i$, by averaging the hidden states in $\mathcal{M}$ for $x_i$ across all layers. This approach for obtaining input representation for consistency calculation has been effectively demonstrated by Atanasova et al. (2020). Let $Dist$ be a distance function; we compute consistency for a dataset $\mathcal{D}$ and model $M$ by measuring Spearman's $\rho$ between similarities in aggregate hidden states ($\mathcal{H}_i$ and $\mathcal{H}_j$) and similarities in attention explanations ($r_i$ and $r_j$) as detailed in eq. 12. We further define our loss function for consistency as the Kullback-Leibler divergence loss between explanations for two samples, weighted by the similarity between the samples' aggregate hidden states. (eq. 13). For additional information the consistency loss function, see §B.4.

$$\text{CONS} =$$
$$\rho\bigg( Dist(\mathcal{H}_i, \mathcal{H}_j), Dist(r_i, r_j) \bigg) \qquad (12)$$

$$\mathcal{L}_{\text{CONS}} =$$
$$\frac{1}{Dist(\mathcal{H}_i, \mathcal{H}_j) + \epsilon_0} \cdot \text{KLDiv}\bigg( r_i, r_j \bigg) \qquad (13)$$

## 3 Experiments

In order to evaluate our framework's effectiveness at producing simulatable, faithful and consistent explanations, we train three transformer-based language models with IvRA: Electra (Clark, 2020), Llama-2-7b (Touvron et al., 2023), and GPT-2 (medium) (Radford et al., 2019)) on three NLP Tasks: Sentiment Classification, Entailment Inference and Question-Answering, with the following datasets, respectively: IMDb (Maas et al., 2011), SNLI(Bowman et al., 2015), and SQuAD (Rajpurkar et al., 2016). In the main paper, we present results of IvRA using ELECTRA as the base language model, with further results using Llama-2 and GPT-2 provided in I. In §3.1, 3.2, 3.3 we report results obtained when training models for each of the interpretability criteria separately. We then delve into mixed-criteria training in §3.4 and examine the IvRA's effect on downstream accuracy in §3.5.

In order to assess the relative effectiveness of IvRA compared to other explanation methods, we conduct experiments using other methods on the same datasets and compare the extent to which each interpretability objective is achieved. We report the mean and standard error values from 5 runs for each experiment setting. The explanation methods that were employed in our experiments are:

- **Common Pooling Techniques**: We obtain explanations by **1**): Averaging the attention distribution over all heads in all layers and **2**): Averaging the attention distribution in heads of the final layer
- **Explainability Methods**: **3)** LIME (Ribeiro et al., 2016), **4)** Input X Gradient (Shrikumar et al., 2016), **5)** Integrated Gradients (Sundararajan et al., 2017)
- **Attention-Regularization**: **5)** Attention-SMaT, the coarsely parameterized attention module introduced by Fernandes et al. (2022). **6)** IvRA with NORM = Softmax **7)** IvRA with NORM = Sparsemax (Martins and Astudillo, 2016)

| | IMDb | SNLI | SQuAD |
|---|---|---|---|
| Attention (Avg. all layers) | $0.911 \pm 0.025$ | $0.906 \pm 0.029$ | $0.821 \pm 0.029$ |
| Attention (last layer) | $0.916 \pm 0.038$ | $0.908 \pm 0.029$ | $0.837 \pm 0.042$ |
| Input X Gradients | $0.827 \pm 0.042$ | $0.813 \pm 0.006$ | $0.773 \pm 0.051$ |
| Integrated Gradients | $0.831 \pm 0.057$ | $0.803 \pm 0.018$ | $0.782 \pm 0.052$ |
| LIME | $0.828 \pm 0.033$ | $0.825 \pm 0.031$ | $0.785 \pm 0.012$ |
| Attention-SMaT | $0.926 \pm 0.035$ | $0.912 \pm 0.013$ | $0.881 \pm 0.043$ |
| IvRA - Softmax | $\underline{0.928} \pm 0.055$ | $\underline{0.922} \pm 0.047$ | $\underline{0.888} \pm 0.028$ |
| IvRA - Sparsemax | $\mathbf{0.944} \pm 0.019$ | $\mathbf{0.939} \pm 0.027$ | $\mathbf{0.897} \pm 0.019$ |

Table 1: Simulatability results of our experiments. Bolded values indicate the highest performance, with underlined values indicating the highest performance.

### 3.1 Simulatability

In Table 1 we show the simulatability accuracy (eq. 6) of our experiments. We observe that, overall, IvRA is more capable of producing simulatable explanations than other methods. We found that the gradient-based explanation methods and LIME did not consistently outperform the common attention-pooling techniques in terms of to simulatability. In addition, we see that using Sparsemax as the normalizing function leads to more simulatable explanations than Softmax. When normalizing with Softmax, all elements are guaranteed a representation in the distribution, however minute it may be.

| Explanations | | Features Heat Map | Heads Heat Map | Layers Heat Map |

**IvRA - Sparsemax**

Amateur, no-budget films can be surprisingly good ...This, however, is not one of them... another Brad Sykes atrocity. The acting is hideous, except for ... who shows some promise.

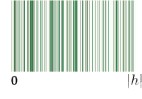
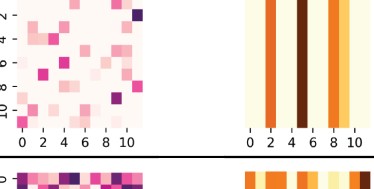

**IvRA - Softmax**

Amateur, no-budget films can be surprisingly good ...This, however, is not one of them... another Brad Sykes atrocity. The acting is hideous, except for ... who shows some promise.

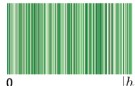
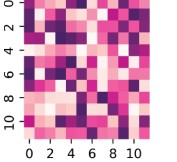
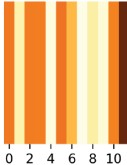

**LIME**

Amateur, no-budget films can be surprisingly good ...This, however, is not one of them... another Brad Sykes atrocity. The acting is hideous, except for ... who shows some promise.

Figure 3: Example explanations and coefficient heat maps from `IvRA` (Softmax and Sparsemax) and LIME. For `IvRA`, a stronger shade denotes a higher importance of that word's influence on the output. For LIME, importance scores are signed, with green and red representing positive influence and negative influence, respectively.

This leads to all tokens always having *some* weight in the explanation. Normalizing with Sparsemax leads to tokens having no weight at all in the explanation, thus producing more sparse and more concise explanations. We show example explanations from LIME (Ribeiro et al., 2016), and `IvRA` when normalizing with Softmax and Sparsemax in Fig. 3, where sparser parameters is observable at all levels when training with Sparsemax than training with Softmax. In addition, we observe `IvRA` is able to produce much concise explanations than LIME. This is intuitive when considered from a human standpoint, as simple and concise explanations are easier to follow along than long-winded explanations.

## 3.2 Faithfulness

We show the comprehensiveness obtained in our experiments in Table 2 and sufficiency scores obtained in our experiments in Table 3. We observe that `IvRA` is able to produce more faithful explanations than methods. We further note that comprehensiveness is the only interpretability criterion in our experiments for which `IvRA`-Softmax consistently outperformed `IvRA`-Sparsemax. We hypothesize that this may be due to the fact that generating explanations with weights distributed over a large number of tokens proves advantageous when assessing comprehensiveness—typically, the more words included in an explanation, the more comprehensive it is. By design, Softmax excels in pro-

ducing explanations that highlight a greater number of tokens. On the other hand, we note that Softmax tends to underperform when used for training aimed at sufficiency. For additional discussion and analysis on the number of words identified and its relationship with faithfulness, please see §D.

| | IMDb | SNLI | SQuAD |
|---|---|---|---|
| Attention (Avg. all layers) | 0.115 ± 0.093 | 0.099 ± 0.106 | 0.018 ± 0.035 |
| Attention (last layer) | 0.115 ± 0.039 | 0.097 ± 0.082 | 0.015 ± 0.051 |
| Input X Gradients | 0.130 ± 0.059 | 0.101 ± 0.094 | 0.021 ± 0.060 |
| Integrated Gradients | 0.148 ± 0.033 | 0.175 ± 0.108 | 0.092 ± 0.011 |
| LIME | 0.139 ± 0.092 | 0.179 ± 0.111 | 0.087 ± 0.065 |
| Attention-SMaT | 0.275 ± 0.253 | 0.350 ± 0.081 | 0.121 ± 0.066 |
| IvRA - Softmax | **0.323** ± 0.080 | **0.427** ± 0.084 | **0.126** ± 0.031 |
| IvRA - Sparsemax | 0.273 ± 0.107 | 0.360 ± 0.061 | 0.111 ± 0.088 |

Table 2: Comprehensiveness results. Bolded values indicate the highest performance, with underlined values indicating the highest performance.

| | IMDb | SNLI | SQuAD |
|---|---|---|---|
| Attention (Avg. all layers) | 0.157 ± 0.051 | 0.514 ± 0.081 | 0.620 ± 0.097 |
| Attention (last layer) | 0.130 ± 0.018 | 0.679 ± 0.066 | 0.728 ± 0.041 |
| Input X Gradients | 0.137 ± 0.019 | 0.487 ± 0.079 | 0.718 ± 0.033 |
| Integrated Gradients | 0.147 ± 0.014 | 0.566 ± 0.021 | 0.622 ± 0.119 |
| LIME | 0.131 ± 0.009 | 0.401 ± 0.038 | 0.531 ± 0.017 |
| Attention-SMaT | 0.129 ± 0.012 | 0.330 ± 0.041 | 0.530 ± 0.012 |
| IvRA - Softmax | 0.132 ± 0.038 | 0.364 ± 0.087 | 0.589 ± 0.032 |
| IvRA - Sparsemax | **0.040** ± 0.030 | **0.220** ± 0.055 | **0.459** ± 0.033 |

Table 3: Sufficiency results. Bolded values indicate the best performance (lowest number), with underlined values indicating the second-best performance.

## 3.3 Consistency

We present our consistency results in Table 4. While `IvRA` clearly outperforms other explainabil-

| | Context | Question/Answer Pairs | Explanation (IG) | Explanation (IvRA - Sparsemax) |

| | Context | Question/Answer Pairs | Explanation (IG) | Explanation (IvRA - Sparsemax) |
|---|---|---|---|---|
| | West was significantly inspired by Roseland NYC Live, a 1998 live album by English trip hop group Portishead, produced with the New York Philharmonic Orchestra. Though West had not been a ble to afford many live instruments around the time of his debut album, the money from his commercial success enabled him to hire a string orchestra for his second album Late Registration. West collaborated with American film score composer Jon Brion, who served as the album's co-executive producer for several tracks. | Question: "What kind of ensemble did Kanye hire to work on his second album?" 

 Answer: string orchestra | West was significantly inspired by Roseland NYC Live, a 1998 live album by English trip hop group Portishead, produced with the New York Philharmonic Orchestra. ... Though West had not been able to afford many live instruments around the time of his debut album, the money from his commercial success enabled him to hire a string orchestra for his second album ... | West was significantly inspired by Roseland NYC Live, a 1998 live album by English trip hop group Portishead, produced with the New York Philharmonic Orchestra. ... Though West had not been a able to afford many live instruments around the time of his debut album, the money from his commercial success enabled him to hire a string orchestra for his second album Late Registration... West collaborated with American film score composer Jon Brion... |
| | | Question: "What composer worked alongside Kanye on the album's production?" 

 Answer: Jon Brion | West was significantly inspired by Roseland NYC Live, a 1998 live album by English trip hop group Portishead, produced with the New York Philharmonic Orchestra. ... West collaborated with American film score composer Jon Brion, who served as the album's co-executive producer for several tracks. ... | West was significantly inspired by Roseland NYC Live, a 1998 live album by English trip hop group Portishead, produced with the New York Philharmonic Orchestra. ... instruments around the time of his debut album, the money from his commercial success enabled him to hire a string orchestra for his second album Late Registration... West collaborated with American film score composer Jon Brion... |

Figure 4: Visualization of explanations for similar instances of data provided by Integrated Gradients and IvRA-sparsemax. We can observe that IvRA-sparsemax is able to produce explanations that are more consistent (highlightin words that are common to both instances of data with similar degrees of emphasis) than IG. For example, the word "instruments" is not highlighted in both instances by IG, whereas IvRA highlights the word with similar emphases in both cases.

| | IMDb | SNLI | SQuAD |
|---|---|---|---|
| Attention (Avg. all layers) | $0.302 \pm 0.208$ | $0.211 \pm 0.291$ | $0.134 \pm 0.032$ |
| Attention (last layer) | $0.312 \pm 0.036$ | $0.138 \pm 0.157$ | $0.194 \pm 0.066$ |
| Input X Gradients | $0.296 \pm 0.088$ | $0.230 \pm 0.244$ | $0.186 \pm 0.084$ |
| Integrated Gradients | $0.319 \pm 0.044$ | $0.232 \pm 0.036$ | $0.146 \pm 0.019$ |
| LIME | $0.338 \pm 0.038$ | $0.273 \pm 0.172$ | $0.224 \pm 0.224$ |
| Attention-SMaT | $0.340 \pm 0.003$ | $0.333 \pm 0.031$ | $0.247 \pm 0.052$ |
| IvRA - Softmax | $\mathbf{0.378} \pm 0.005$ | $\underline{0.336} \pm 0.052$ | $\underline{0.250} \pm 0.14$ |
| IvRA - Sparsemax | $\underline{0.366} \pm 0.037$ | $\mathbf{0.357} \pm 0.042$ | $\mathbf{0.284} \pm 0.07$ |

Table 4: Consistency results. Bolded values indicate the highest performance, with underlined values indicating the highest performance.

ity methods, we do not observe a clear winner between IvRA-Softmax and IvRA-Sparsemax. In particular, we observe overlapping IQR's between SMaT, IvRA-Softmax and IvRA-Sparsemax. In Fig. 4, we show the explanations produced by Integrated Gradients and IvRA-sparsemax's for two instances of data from SQuAD with similar semantics. In the example, both questions inquire about collaborators with whom Kanye West has previously worked on his album. While IG's explanation is sporadic and pattern-less (e.g. 'inspired' having completely opposite color/weight of contribution in the two examples), we observe consistent highlighting of keywords by IvRA-Sparsemax in both examples.

### 3.4 Mixed-criteria Training

To what extent does the simulatability training of a model correlate with its comprehensiveness? Can a model trained to produce sufficient explanations also be consistent? These questions arise from the multifaceted nature of IvRA, which aims to accommodate various interpretability criteria. While earlier sections demonstrate IvRA's superiority over existing methods when trained individually for each criterion, this section explores the efficacy of mixed-criteria training. We train IvRA with dif-

ferent combinations of interpretability losses (Eqn. 7, 10, 11, 13) enabled. We then compare the results of these models against models that were trained solely using each individual criterion. Formally, let $\mathcal{C} = $ SIM, COMP, SUFF, CONS, and $\mathcal{P}(C)$ denote the powerset of $\mathcal{C}$. To assess the effectiveness of mixed-criteria training, we evaluate the Average Relative Gain (ARG) (Ye et al., 2021) of criterion $a \in \mathcal{C}$ for an IvRA model trained on $b \in \mathcal{P}(\mathcal{C})$ against a model trained solely on $a$. Our findings are presented in a heatmap shown in Fig. 5. We observe underperformance (negative ARG) across-the-board i.e., we observe that models trained with multiple criteria losses enabled achieve each criterion *less* than models trained with a sole focus on the same criterion. While this may initially seem discouraging, we find the results to be intuitive—as we demonstrate, models trained to be more simulatable do not naturally exhibit greater comprehensiveness than models originally trained with comprehensiveness as the primary goal. Moreover, our results suggest that different interpretability criteria's parameters are at odds with each other and satisfying one criterion may not satisfy others. We observe that, particularly, training for consistency has the greatest adverse effect on all other criteria. We also observe the least amount of decrease in performance between models trained for simulatability and sufficiency and vice versa. We attribute this to our earlier discussions in §3 where we find that conciser explanations are helpful for both simulatability and sufficiency.

### 3.5 Impact on Downstream Accuracy

In this section, we examine IvRA's influence on model performance. We specifically aim to assess how each of the four interpretability criteria

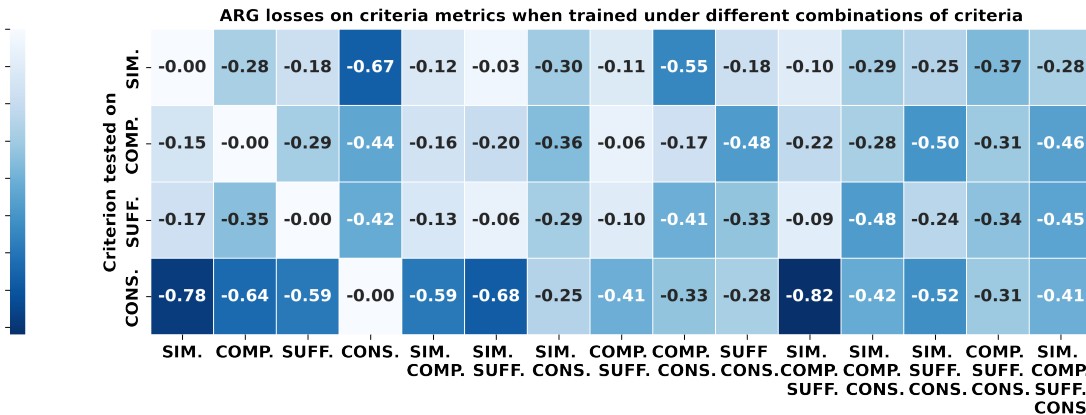

Figure 5: Average Relative Gain (ARG) when IvRA is trained under combinations of interpretability criteria (x-axis) over when IvRA is trained individually for each criterion (y-axis). In general, we observe CONS. to be most at odds with other interpretability criteria. We also observe SIM. and SUFF. to be the most compatible pair of criteria.

| Inter. criteria enabled | IMDb | SNLI | SQuAD |
|---|---|---|---|
| {Sim.} | 95.0 (-0.4) | 89.8 (-1.4) | 89.0 (-0.9) |
| {Comp.} | 93.1 (-2.3) | 87.5 (-3.7) | 88.0 (-1.9) |
| {Suff.} | 95.2 (-0.2) | 90.3 (-0.9) | 89.4 (-0.5) |
| {Cons.} | 87.6 (-7.8) | 84.7 (-6.5) | 79.5 (-10.4) |
| {Sim., Comp.} | 93.6 (-1.8) | 88.7 (-2.5) | 88.7 (-1.2) |
| {Sim., Suff.} | 95.2 (-0.2) | 90.0 (-1.2) | 89.2 (-0.7) |
| {Sim., Cons.} | 90.0 (-5.4) | 86.8 (-4.4) | 81.2 (-8.7) |
| {Comp., Suff.} | 94.3 (-1.1) | 88.4 (-2.8) | 88.8 (-1.1) |
| {Comp., Cons.} | 88.8 (-6.6) | 85.9 (-5.3) | 82.0 (-7.9) |
| {Suff., Cons.} | 91.1 (-4.3) | 87.7 (-3.5) | 82.8 (-7.1) |
| {Sim., Comp., Suff., Cons.} | 91.8 (-3.6) | 88.5 (-2.7) | 83.1 (-6.8) |
| ELECTRA baseline | 95.4 | 91.2 | 89.9 |

Table 5: The accuracy of IvRA when training with different interpretability criteria (Sim., Comp., Suff. Cons.) enabled. Shades of blue indicate the relative decrease in accuracy when comapred against the baseline model, with lighter shades indicating smaller decreases and darker shades indicating larger decreases in accuracy.

in IvRA impacts downstream accuracy. To that end, we conduct experiments on accuracy by varying the combination of loss functions used during training (Eqn. 7, 10, 11, 13), while always including Cross-entropy loss. We present our results on model accuracy across three three NLP tasks in Table. 5. Despite some decreases in accuracy, IvRA's effect on accuracy is generally minor, except in cases involving consistency training. Furthermore, distinct impacts on accuracy are observed across the four interpretability criteria. Notably, training for Sim. and Suff. demonstrates minimal accuracy reduction. We hypothesize, akin to sections §3.1 and §3.2, that training for these criteria involve pinpointing salient and succinct input elements, aligning well with accuracy training. Comp. training also involves identifying salient features but with

less emphasis on succinctness, which we believe is the reason for a slightly higher decrease in accuracy. On the other hand, Cons. training causes a relatively large decrease in model accuracy because its goal of identifying similar elements in similar inputs lacks a direct alignment with the accuracy (Cross-entropy) objective compared to other criteria. Overall, training an IvRA model with any combination of criteria from Table 5 yields models with competent downstream accuracy. We consider IvRA's reliable performance across various criteria combinations as evidence of its robustness, balancing specificity in producing interpretable explanations with the generalizability required for accurate predictions. We further explore IvRA's generalizability in §E.

## 4 Conclusion

We introduce IvRA, a paramerterized attention module for directly training a LM's attention distribution to produce explanations that align with interpretability criteria. We test IvRA's effectiveness at producing explanations that are simulatable, faithful (comprehensive and sufficient) and consistent using multiple LMs and on multiple NLP tasks. We perform ablation experiments to reveal insights on the interplay between different interpretability criteria and to assess IvRA's influence on downstream accuracy. Our findings demonstrate that IvRA's attention-based explanations is robust under various settings and empowers LMs to generate explanations that better align with interpretability criteria.

## 5 Limitation and Future Direction

We summarize the main limitations of our work below. While we acknowledge the potential shortcomings of this work in these areas, we also hope to inspire future works of research in these areas to address and improve upon our deficiencies.

1. **Reliance on Existing Interpretability Metrics:** Our method builds upon existing interpretability metrics like faithfulness, consistency, and simulatability. Despite their widespread use, these metrics may not fully capture the complexity of interpretability in machine learning models. Developing more comprehensive and robust metrics could potentially enhance our approach and lead to better results

2. **Generalizability:** The performance of our proposed method is primarily assessed on specific datasets and tasks. Thus, its applicability and effectiveness across different domains, tasks, and model architectures remain to be further explored

3. **Scalability:** Our method relies on the introduction of additional loss functions and the training of student models, which might introduce computational overhead and increase training complexity

4. **Subjectivity of Interpretability:** Interpretability is inherently subjective, and what might be interpretable for one user or expert may not necessarily be so for another. Our work focuses on commonly used metrics and techniques, which may not capture diverse perspectives on interpretability. Developing adaptive and *specialized* interpretability approaches could be a valuable direction for future research .

## Acknowledgements

This research was supported in part by grants from the US National Library of Medicine (R01LM012837 & R01LM013833), the US National Cancer Institute (R01CA249758), the US National Science Foundation (NSF Award 2242072), and the John Templeton Foundation. Additionally, we would like to express our sincerest gratitude to Naofumi Tomita, Joseph DiPalma, Alex DeJournett, and Richard Holcomb IV for their help and support during the research stage and the writing of this paper.

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

# A  Additional Discussion on Related Works

## A.1  Attention as Explanation and Regularized Attention as a Technique

Although recent studies like Singh et al. (2022); Jie et al. (2024); Yang et al. (2024) have explored prompting-based approaches to explain LLMs' decisions, attribution-based methods that leverage attention scores remain the dominant and conventional techniques for interpretability. Seminal efforts on attention as an explanation (Serrano and Smith, 2019; Jain and Wallace, 2019), have focused on assessing the quality of explanation along axioms such as consistency, but they do not extend to training a model's attention weights to enhance these qualities. More recent works on evaluating extracted attention rationales (DeYoung et al., 2019; Atanasova et al., 2020) have primarily scrutinized the **post-hoc explanation methods'** abilities to align with faithfulness and consistency. While

Chan et al. (2022b) proposed a framework to optimize a model for task loss and faithfulness, it relies on a separate rationale extractor model. To directly impact the interpretability of transformer models, recent work has proposed attention-regulating techniques that adjust a model's attention weights to generate more interpretable explanations (Treviso and Martins, 2020; Guerreiro and Martins, 2021). However, Treviso and Martins (2020) only explored the effectiveness of sparse attention as a communication method to a lay person. Similarly, Ferreira et al. (2024) examines the capability of attribution methods to produce interpretable explanations, but only on sentiment classification tasks and only on human judgement-based criteria such as plausibility. On the other hand, although Meister et al. (2021) found that sparse attention does not necessarily produce *plausible* explanations, they did not evaluate sparse attention using the *other* interpretability criteria outlined in our paper. Additionally, Meister et al. (2021) suggests that future research could explore interpretability experiments on attention outputs using the evaluation strategies of DeYoung et al. (2019), which we have incorporated through our adaptations of comprehensiveness and sufficiency.

## A.2 Defining and Evaluating Interpretability Objectives

Doshi-Velez and Kim (2017) defined forward simulation of model decisions by humans as a core interpretability metric. Pruthi et al. (2022) then extended the evaluation of simulatability with an automated setup that relied on simulating with models instead of humans. For faithfulness, DeYoung et al. (2019) introduced concrete measures in the form of comprehensiveness and sufficiency. Consistency was discussed in Jain and Wallace (2019); Serrano and Smith (2019). More recently, Atanasova et al. (2020) benchmarked the consistency of explainer on several datasets and found that attention-based explainer generally outperformed gradient-based explainer in terms of consistency. Neely et al. (2021) shows that without comparison against ground-truth explanations (often provided by human-labeled rationales (DeYoung et al., 2019)), it is difficult to establish an objective better/worse explainer. Even more recent work (Joshi et al., 2022; Chan et al., 2022a) show that it is difficult to align attention networks' output with *human* rationales (*plausibiltiy*). In terms

of the architecture used to evaluate interpretability, Guerreiro and Martins (2021), Jacovi and Goldberg (2021) and Ismail et al. (2021) take the approach of building model decisions upon aligned rationales, but focus on task performance and evaluate their work only on a subset of the interpretability objectives.

## A.3 Where our work stands

IvRA as a an explanation technique that utilizes regularized attention has the advantage over gradient and perturbation-based methods (Ribeiro et al., 2016; Shrikumar et al., 2017) in that the process of explaining the output is intrinsic to the model and not decoupled from the prediction process. In addition, IvRA does not require the usage of a separate model or gradient-based salience explainer to act as rationale extractor during training, as in the case of Chan et al. (2022b) and Ismail et al. (2021). Moreover, IvRA's is demonstrably robust for a wide range of interpretability criteria (simulatability, comprehensiveness, sufficiency and consistency) whereas techniques in Fernandes et al. (2022), Chan et al. (2022b), Xie et al. (2022b) and Ismail et al. (2021) have only be shown to be effective at enhancing model interpretability for a *subset* of criteria. Finally, our work's scope is similar to that of Sun et al. (2024), as both attempt to create a comprehensive framework for evaluating different interpretability criteria. However, while their work emphasizes diagnosing the properties of *existing* interpretability techniques, ours is focused on methods to *train* models to *acquire* these properties. Furthermore, we affirm the validity of our approach by highlighting that several of our interpretability criteria are closely aligned with those defined by Sun et al. (2024).

## B Interpretability Criteria Details

## B.1 Simulatability Training Details

We use a **fine-tuned** model for $\mathcal{M}$ trained on the dataset for the task and an **unfine-tuned** model for $\mathcal{S}$ that has not been exposed to the dataset. Additionally, $\mathcal{M}$'s explanations ($r_i$'s) are withheld from the student $\mathcal{S}$ during testing to prevent information leakage (Pruthi et al., 2022).

## B.2 Gradient updates

Optimizing the $\mathcal{M}$'s attention parameters $\theta(\mathcal{M})$ for the simulatability of a separate, student model $\mathcal{S}$ is a non-trivial process. We in this work take

the *scaffoled* approach for optimizing the parameters of introduced in by Fernandes et al. (2022). Specifically, let $d \in \mathcal{D}$ be a batch of data, we frame optimizing IvRA's attention weights as a bi-level optimization problem where eq. 14 updates $\theta(\mathcal{S})$ ($\mathcal{S}$'s parameters) based on model outputs (see eq. 7) and eq. 15 updates $\mathcal{M}$'s parameters based on how well $\mathcal{S}$ can simulate $\mathcal{M}$, with newly updated parameters. We take a single optimization step to calculate the gradient for $\mathcal{S}$ (eq. 14). After updating $\mathcal{S}$ with the gradient, we take an *additional* gradient step but only use this gradient to update parameters of $\mathcal{M}$, and not $\mathcal{S}$ (eq. 15). For even more specific details on scaffolded simulatability, we refer the reader to *pilot updates* by Zhou et al. (2021).

$$\theta^*(\mathcal{S}) = $$
$$\mathbf{argmin}_{\theta(\mathcal{S})} \; \mathbb{E}\left\{ \mathcal{L}_{\mathsf{SIM}}[d, \mathcal{M}, \mathcal{S}] \right\} \quad (14)$$

$$\theta^*(\mathcal{M}) = $$
$$\mathbf{argmin}_{\theta(\mathcal{M})} \; \mathbb{E}\left\{ \mathcal{L}_{\mathsf{SIM}}[d, \mathcal{M}, \mathcal{S}_{\theta^*(\mathcal{S})}] \right\} \quad (15)$$

### B.2.1 Constrained student training

In addition to limiting the model weights of $\mathcal{M}$ and $\mathcal{S}$ as described in §2.2.1, we further constrain the amount of data the student $\mathcal{S}$ is trained on. Specifically, while the full training set for IMDb, SNLI and SQuAD were used to finetune and train $\mathcal{M}$, we only use 20% of the available testing set to train the student, which yields 5000 samples, 2000 samples and 2000 samples for IMDb, SNLI and SQuAD, respectively.

### B.3 Faithfulness Training Details

### B.3.1 Comprehensiveness and sufficiency bounds

Intuitively, the entropy (with respect to the output class $\tilde{y}_i$) can be higher when calculated with $t_i^k$ removed than when calculated with the entirety of $x_i$, i.e. $-\tilde{y}_i\log(\mathcal{M}(x_i \backslash t_i^k)) \gg -\tilde{y}_i\log(\mathcal{M}(x_i))$. Without bounding by $\mu_{\mathsf{COMP}}$, eq. 10 can yield large, negative losses. While this can analogously happen for eq. 11, there exists alternative loss functions (see below). We experimented with $\mu_{\mathsf{COMP}} \in \{0.1, 0.2, ..., 1\}$ and $\mu_{\mathsf{SUFF}} \in \{0.1, 0.2, ...1\}$ and the reported results in Table 2 and Table 3 are for $\mu_{\mathsf{COMP}} = 1$ and $\mu_{\mathsf{SUFF}} = 0.1$, respectively.

### B.3.2 Sufficiency losses

Apart from the loss function outlined in §2.2.2, we experiment with two additional sufficiency loss functions for sufficiency. Critically, we note that eq. 16 relies on the assumption that $\mathcal{M}$ is not able to make more accurate predictions when using only a subset of the sequence $(t_i^k)$ as its input. We also note here that while 11 and 16 are computed with respect to the output class $\tilde{y}_i$, 17 computes the KL divergence loss over distributions. Similar to (Chan et al., 2022b), we found that all three loss functions can be used to train IvRA for sufficiency, although we decided to report 11 in the main paper as it's more general and in conformity with eq. 10.

$$\mathcal{L}_{\mathsf{MAE-SUFF}} = $$
$$\left| -\tilde{y}_i\log(\mathcal{M}(t_i^k)) + \tilde{y}_i\log(\mathcal{M}(x_i)) \right| \quad (16)$$

$$\mathcal{L}_{\mathsf{KL-SUFF}} = \mathrm{KLDiv}\left( \mathcal{M}(t_i^k), \; \mathcal{M}(x_i) \right) \quad (17)$$

### B.4 Consistency Training Details

### B.4.1 Consistency within batch

For simplicity and clarity, we defined eq. 12 and eq. 13 in the main paper for two examples $x_i$ and $x_j$. In practice, both CONS and $\mathcal{L}_{\mathsf{CONS}}$ are calculated for every pair of samples within each batch during training. We report the consistency for a dataset by averaging the consistency across batches. i.e. CONS we calculate:

$$\frac{1}{|\mathcal{D}|} \frac{1}{|\binom{d}{2}|} \sum_{d \in \mathcal{D}} \sum_{(x_i, x_j) \in \binom{d}{2}} \mathsf{CONS}(x_i, x_j) \quad (18)$$

For batch loss $\mathcal{L}_{\mathsf{CONS}}$ during training, we calculate the following:

$$\frac{1}{|\binom{d}{2}|} \sum_{(x_i, x_j) \in \binom{d}{2}} \mathcal{L}_{\mathsf{CONS}}(x_i, x_j) \quad (19)$$

As a result, we note here that training for pair-wise consistency can be costly in terms of time. For more analysis on computational cost of IvRA see §G.

### B.4.2 Distance function

We report results for using L2 distance as our $Dist$ function. Although we experimented with L1 distance function, we found using L2 distance in general led to better performance. We note here that

the $Dist$ function in eq. 12 calculates pair-wise distance at the *token* level, whereas the $Dist$ function for loss calculate (eq. 13) is the p-2 norm of the difference between $\mathcal{H}_i$ and $\mathcal{H}_j$.

### B.4.3 Consistency clustering loss

Training with a focus on consistent reasoning shares similarities with the process of clustering similar examples together. To that end, we also experimented with a clustering loss for $\mathcal{L}_{\mathsf{CONS}}$ that is similar to the loss function for learning associations between examples in Chen et al. (2019) and Das et al. (2022). Specifically, let $\chi$ and be the set of samples in $d$ that belong belong to the same output class as $x_i$, let $\gamma$ be the set of samples in $d$ that **do not** belong to the same output class as $x_i$ i.e. $\chi = \{x_j \text{ s.t. } y_j = y_i \mid \forall x_j \in d\}$ and $\gamma = \{x_j \text{ s.t. } y_j \neq y_i \mid \forall x_j \in d\}$, we define an alternative clustering loss for consistency training as:

$$
\begin{aligned}
\mathcal{L}_{\mathsf{CLUST-CONS}} = \\
\frac{1}{|\chi|}\sum_{x_j \in \chi}\mathbf{min}||r_i - r_j||_2^2 \\
+ \\
\frac{1}{|\gamma|}\sum_{x_j \in \gamma}\mathbf{max}||r_i - r_j||_2^2
\end{aligned}
\tag{20}
$$

Intuitively, we try to train for consistency via minimizing the distance of $r_i$ and $r_j$'s that are explanations of examples with the same class as $x_i$ and maximizing the distance between $r_i$ and $r_j$'s that are explanations of examples with a different class than $x_i$. In practice, we found this loss function to perform worse than eq. 13 both in terms of consistency as well as time.

### C Effect of Feature/Head/Layer Selection

The interpretable attention module of `IvRA` involves the selection of salient input elements at three levels: feature, head, and layer. What is the effectiveness of the selection process at each level in terms of achieving simulatable, comprehensive, sufficient, and consistent explanations? In this section, we conduct experiments aimed at answering this question. Specifically, we experiment with `IvRA` by enabling feature-level selection, head-level selection, and layer-level selection separately to observe their individual effects during training. The loss curve for each criterion during

training is illustrated in Figure. 6. We observe in our experiments, that, across all four interpretability criteria, layer-level selection exhibits the least reduction in loss during training. While head-level selection is shown to be more effective than layer-level selection, its loss curve stabilizes at a higher level compared to feature-level selection. Notably, feature-level selection proves to be the most effective (out of the three levels) in identifying information that aligns with each of the interpretability criteria, leading to the lowest level of losses during training, relatively to head and layer-level selection. Finally, training with selection at all levels enabled proves to be the optimal solution to produce explanations that align with each of the criteria, albeit with only marginal improvements over feature-selection-only in certain cases.

### D Important Tokens Identified

In Fig. 7, we conduct an analysis of the number of *important* tokens in the output of different explainers. Every token receives *some* weight in the saliency outputs by Integrated Gradients (IG), LIME, and Softmax, although often minute. To find impactful tokens, we perform min-max normalizing on the saliency outputs of these explainers and find the number of tokens (as a percentage of the input's length) that score above thresholds in the set $Z = \{0.1, 0.2, ..., 0.9\}$. i.e. a token is *important* if its normalized saliency is higher than $z \in Z$. We then calculate the area-over-precision curve (DeYoung et al., 2019; Xie et al., 2023) $\forall z \in Z$ to obtain the AOPC of important words identified. We find that, while the number of important remains roughly the same for `IvRA`-Softmax when trained on both COMP. and SUFF, `IvRA`-Sparsemax, in general, identifies fewer tokens when trained for SUFF than when trained for COMP.

### E Transferability between Datasets

We hypothesize that the parameters learned by `IvRA` are transferable between datasets for the same task. To verify our hypothesis, we take models that were trained on IMDb and SNLI, denoted as $\mathcal{M}_I$ and $\mathcal{M}_S$, respectively, and apply them on SST2 and MNLI from GLUE (Wang et al., 2018). In order to gauge the transferability, we directly train another set of models on SST2 and MNLI, denoted as $\mathcal{M}_I^*$ and $\mathcal{M}_S^*$, respectively. We then compare the results of $\mathcal{M}_I^*$ and $\mathcal{M}_S^*$ against the results of $\mathcal{M}_I$ and $\mathcal{M}_S$ using ARG. We report the ARG of

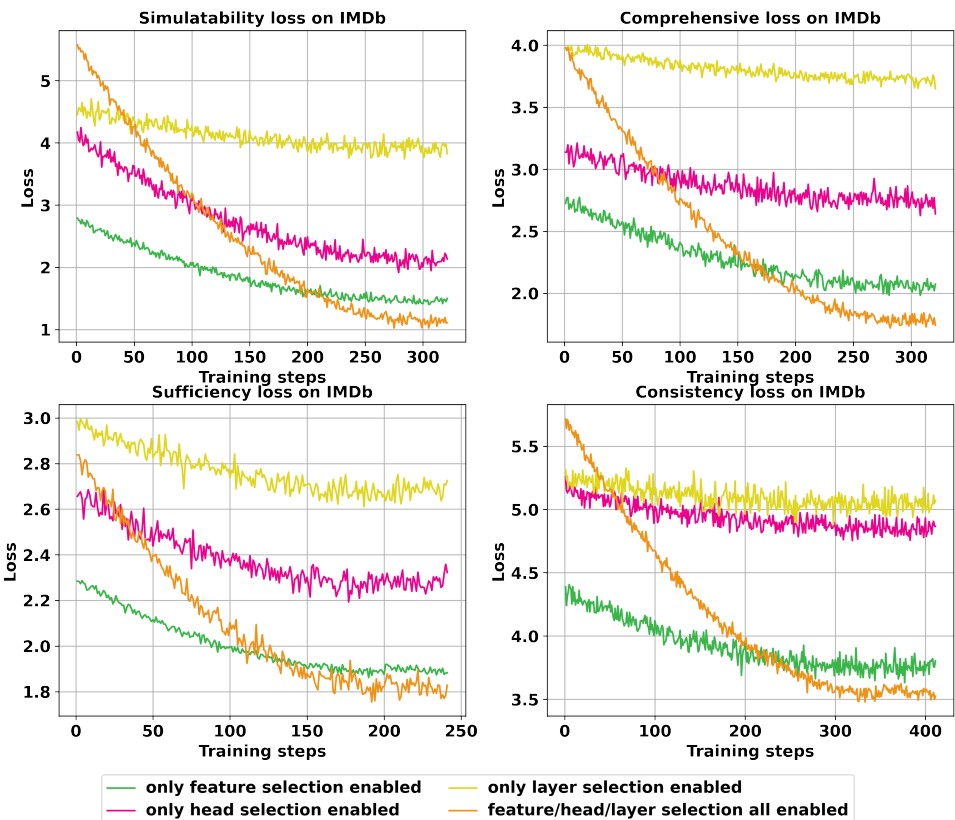

Figure 6: Loss curves for interpretability criteria during training with feature/head/layer-level selection enabled in interpretable attention modules.

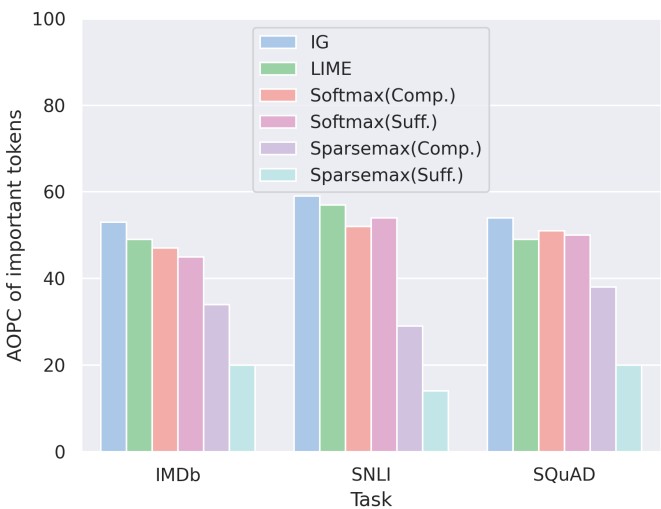

Figure 7: AOPC of important tokens identified by different explainers in different tasks. We observe that IG, in general, identifies the most amount of tokens while training IvRA while normalizing with sparsemax yields the least amount of tokens.

$\mathcal{M}_I^*$ and $\mathcal{M}_S^*$ over $\mathcal{M}_I$ and $\mathcal{M}_S$ in percentages in Fig. 8. A higher ARG means a greater difference in scores for each interpretability criterion between the directly-trained models and the models with transferred parameters. We observe that the parameters trained for SIM transferred the best

between datasets, followed by SUFF and COMP. We also note that parameters trained for CONS did not transfer well, relatively speaking. We conjecture that, although the task for both datasets are the same, the difference in the semantics of samples between two datasets can vary widely, thus making

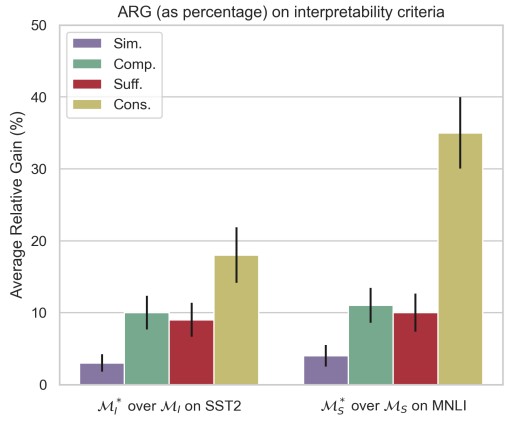

Figure 8: Average Relative Gain (in terms of SIM, COMP, SUFF and CONS) of IvRA-Sparsemax (Llama-2) when trained on SST2/MNLI over when trained on IMDB/SNLI.

it difficult for CONS parameters learned on one dataset to be applied to another.

## F    The Plausibility of IvRA

Plausibility is defined as how convincing are explanations to humans (Jacovi and Goldberg, 2020). Recent studies have assessed plausibility by measuring the overlap between generated rationales (a set of tokens) and groundtruth labels (Sun et al., 2024; Xie et al., 2022a). However, this approach not only highlights that plausibility is inherently human judgement-based and challenging to train for—requiring a distinct set of labeled groundtruth data for each dataset (Chan et al., 2022b)—but also that recent research suggests attributions might be ineffective in producing plausible outputs altogether (Ferreira et al., 2024). Consequently, in this work, we have chosen *not* to use plausibility as a training objective. Nonetheless, we include a study on the plausibility of explanations generated by IvRA models trained under *alternative* criteria here.

Similarly to simulatability, plausibility derives its advantage and utility as an evaluation metric from its alignment with human intuition. Therefore, we in this study conduct a plausibility study exploring which of IvRA models provides the most plausible explanations. We utilize the annotated MovieReviews dataset (DeYoung et al., 2019) which consists of human-labeled rationales for movie review sentiment classification. The rationales are in the form of tokens that have binary

labels 0 and 1 that indicate their presence in the rationale. For each of the explanation method in Table 6, we calculate the plausibility score as the AUC ROC of tokens identified against salient tokens labeled by human annotators. We found the explanations generated by IvRA to be the most plausible i.e., aligning the most with human-generated rationales in terms of tokens identified. More specifically, we find that the explanations learned for the criterion of simulatability are the most plausible overall, followed by sufficiency, comprehensiveness and consistency. This study, in conjunction with our findings in §3.1, show us that sparser explanations that can better target keywords are deemed more intuitive and practical by both models and humans alike. Additionally, we observe , apart from IvRA(Cons.), explanations produced by models incorporating learnable interpretable attention modules (IvRA(SIM., COMP, SUFF) & SMaT) outperformed perturbation and gradient-based methods such as LIME and IG in generating more plausible explanations.

## G    Computational Cost of IvRA

This section explores the computational overhead associated with training and deploying IvRA to generate explanations. To assess the time complexity of IvRA during training, we employ an IvRA model (utilizing Llama-2 as the base) for each interpretability criterion using varying quantities of input data. Specifically, we conduct training for 10 epochs with $N \in 10, 100, 1000, 10000, 100000$ input samples from the SNLI dataset and measure the elapsed time in minutes. The outcomes of these experiments, depicted in Figure. 9, reveal that IvRA introduces only a marginal increase in training time complexity compared to the baseline model[3]. It is important to note that training IvRA for all criteria except consistency proves to be feasible in terms of time. Furthermore, even in the case of consistency, the training time only becomes computationally challenging for input samples of very large sizes ($N \geq 100000$).

In terms of explanation generation time, IvRA presents a distinct advantage over existing post-hoc explanation methods like Ribeiro et al. (2016) and Shrikumar et al. (2016, 2017). Unlike gradient-based post-hoc techniques, IvRA does not necessitate gradient calculations during inference, thereby

---

[3]Details regarding our computational hardware are outlined in §H

| Explanation Methods | Plausibility |
|---|---|
| Attention (Avg.) | $0.68 \pm 0.03$ |
| Attention (Last layer) | $0.61 \pm 0.02$ |
| Input X Gradients | $0.53 \pm 0.03$ |
| Integrated Gradients | $0.51 \pm 0.02$ |
| LIME | $0.58 \pm 0.04$ |
| SMaT | $0.73 \pm 0.02$ |
| IvRA Sparsemax (trained for SIM.) | $\mathbf{0.78 \pm 0.03}$ |
| IvRA Sparsemax (trained for COMP.) | $0.62 \pm 0.04$ |
| IvRA Sparsemax (trained for SUFF.) | $0.72 \pm 0.03$ |
| IvRA Sparsemax (trained for CONS.) | $0.53 \pm 0.06$ |

Table 6: The plausibility score (as AUC ROC of identifed tokens) of XAI methods and IvRA on the MovieReviews dataset. The IvRA Sparsemax trained for simulatability is shown to produce the most plausible explanations over all other explanation methods.

reducing computational complexity during its application. We conducted experiments comparing the time required for popular post-hoc methods and `IvRA` to generate explanations across different input sample sizes, as depicted in Figure. 10. Our results indicate that, for all versions of `IvRA`, the time needed to generate explanations is significantly shorter compared to post-hoc methods. Overall, while deploying a `IvRA` model may involve additional time complexity during the training phase, we found this to be manageable in implementation. Furthermore, `IvRA` offers the added benefit of producing superior (more simulatable, faithful, and consistent) explanations at faster speeds during application.

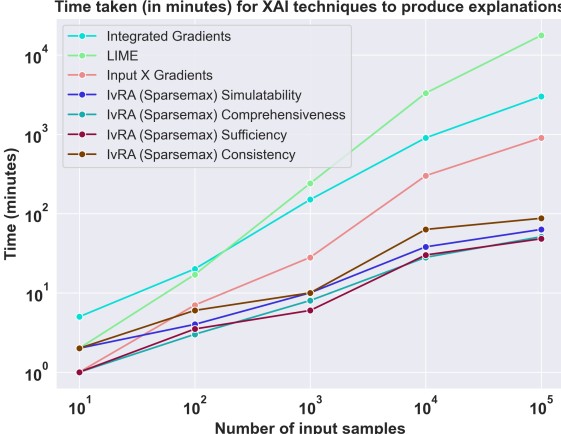

Figure 10: Time required by XAI methods and `IvRA` to generate explanations for different input sizes. `IvRA` exhibits notable advantages over alternative post-hoc XAI methods, especially noticeable with larger input sizes.

## H Compute resources and Additional Hyperparameters

Our compute resources consist of $4\times$ RTX 6000, $4\times$ RTX 4500 and $2\times$ RTX 3090. For running Integrated Gradients in our experiments, we use 50 iterations for calculating the integral. For running LIME in our experiments, we use 500 perturbations to approximate the neighborhood in which the surrogate models are learned. For baseline embeddings, we use zero tensors (Atanasova et al., 2020). Saliency scores (for each individual word) in all settings are the sum of saliency scores of its word pieces (DeYoung et al., 2019). We use AdamW (Loshchilov and Hutter, 2017) as our optimizer for all our models, with the exception of training the student for simulatability, in which case we use

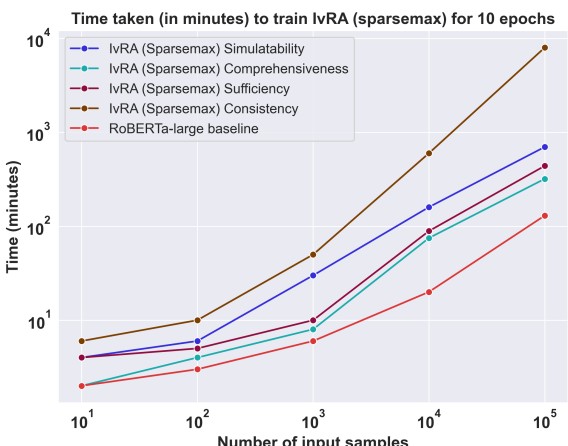

Figure 9: Growth in training time for `IvRA` with respect to input sample size. While employing `IvRA` does introduce a slight increase in training time, this additional time is generally manageable for most criteria, except when training for consistency with very large sample sizes.

SGD (§B.1).

In all the experiments detailed in Tables 1, 2, 3, and 4, we trained all models for 20 epochs at a batch size of 64, applying decay factor every two epochs, and the reported results are from the best iteration of each model. To determine the optimal learning rate, we explored a broad range of learning rates and decay factors. The outcomes of our hyperparameter search for the IMDb task are presented in Table 7. Our investigation reveals that while IvRA can acquire the necessary information for producing interpretable explanations across most settings, achieving the **optimal** performance metrics requires specific learning rates, highlighting IvRA's sensitivity to variations in learning rates. In the experiments documented in Tables 1, 2, We observe that training for simulatability achieves optimal results with smaller learning rates and higher decay factors (larger $\gamma$). Both comprehensiveness and sufficiency training benefit from a moderate learning rate and decay factor. Training for consistency performs best with a higher initial learning rate and a lower decay factor.

## I  Additional Experimental Results

| LR | $\gamma$ | SIM.↑ | COMP.↑ | SUFF.↓ | CONS.↑ |
|---|---|---|---|---|---|
| | 1.000 | $92.03 \pm 0.75$ | $0.301 \pm 0.118$ | $0.132 \pm 0.110$ | $0.238 \pm 0.109$ |
| 3e-5 | 0.750 | $\mathbf{94.31} \pm 0.35$ | $0.289 \pm 0.102$ | $0.185 \pm 0.115$ | $0.220 \pm 0.111$ |
| | 0.500 | $93.31 \pm 0.45$ | $0.320 \pm 0.124$ | $0.111 \pm 0.109$ | $0.232 \pm 0.107$ |
| | 0.250 | $93.04 \pm 0.92$ | $0.298 \pm 0.155$ | $0.129 \pm 0.014$ | $0.329 \pm 0.110$ |
| | 1.000 | $89.03 \pm 0.50$ | $0.265 \pm 0.088$ | $0.123 \pm 0.051$ | $0.329 \pm 0.110$ |
| 3e-4 | 0.750 | $88.02 \pm 0.84$ | $0.278 \pm 0.096$ | $0.052 \pm 0.053$ | $0.376 \pm 0.108$ |
| | 0.500 | $86.02 \pm 0.88$ | $\mathbf{0.325} \pm 0.068$ | $\mathbf{0.016} \pm 0.013$ | $0.381 \pm 0.108$ |
| | 0.250 | $84.23 \pm 1.20$ | $0.305 \pm 0.083$ | $0.037 \pm 0.012$ | $0.398 \pm 0.109$ |
| | 1.000 | $89.13 \pm 1.80$ | $0.205 \pm 0.077$ | $0.123 \pm 0.044$ | $0.428 \pm 0.109$ |
| 1e-4 | 0.750 | $83.02 \pm 1.94$ | $0.228 \pm 0.121$ | $0.116 \pm 0.013$ | $0.426 \pm 0.111$ |
| | 0.500 | $83.31 \pm 1.37$ | $0.233 \pm 0.098$ | $0.230 \pm 0.009$ | $0.402 \pm 0.109$ |
| | 0.250 | $80.74 \pm 1.34$ | $0.258 \pm 0.078$ | $0.097 \pm 0.011$ | $\mathbf{0.430} \pm 0.108$ |
| | 1.000 | $65.03 \pm 2.80$ | $0.302 \pm 0.076$ | $0.147 \pm 0.082$ | $0.421 \pm 0.108$ |
| 1e-3 | 0.750 | $71.02 \pm 1.94$ | $0.260 \pm 0.132$ | $0.253 \pm 0.057$ | $0.347 \pm 0.110$ |
| | 0.500 | $74.31 \pm 1.37$ | $0.287 \pm 0.079$ | $0.270 \pm 0.034$ | $0.399 \pm 0.109$ |
| | 0.250 | $75.03 \pm 1.34$ | $0.280 \pm 0.109$ | $0.278 \pm 0.011$ | $0.407 \pm 0.107$ |

Table 7: Metrics of interpretability criteria achieved by IvRA with Llama-2 when trained under different learning rates and weight decay factor ($\gamma$) on IMDb. Results ($\mu \pm \sigma$) were obtained from 5 separate runs. Optimal performances for each criterion is **bolded**.

| | | IMDb | SNLI | SQuAD |
|---|---|---|---|---|
| | Attention (Avg. all layers) | $90.46 \pm 0.22$ | $90.88 \pm 0.16$ | $82.18 \pm 0.18$ |
| | Attention (last layer) | $90.65 \pm 0.23$ | $90.45 \pm 0.36$ | $83.62 \pm 0.33$ |
| | Input X Gradients | $82.04 \pm 0.39$ | $80.89 \pm 0.21$ | $76.68 \pm 0.29$ |
| GPT-2 | Integrated Gradients | $82.41 \pm 0.15$ | $79.34 \pm 0.17$ | $78.20 \pm 0.15$ |
| | LIME | $82.16 \pm 0.28$ | $82.45 \pm 0.29$ | $77.30 \pm 0.09$ |
| | Attention-SMaT | $92.53 \pm 0.32$ | $\underline{91.70} \pm 0.15$ | $88.11 \pm 0.28$ |
| | IvRA - Softmax | $\underline{92.09} \pm 0.41$ | $91.21 \pm 0.19$ | $\underline{87.73} \pm 0.46$ |
| | IvRA - Sparsemax | $\mathbf{93.60} \pm 0.42$ | $\mathbf{93.55} \pm 0.43$ | $\mathbf{88.22} \pm 0.34$ |
| | Attention (Avg. all layers) | $91.00 \pm 0.08$ | $90.80 \pm 0.23$ | $82.34 \pm 0.21$ |
| | Attention (last layer) | $91.43 \pm 0.43$ | $91.12 \pm 0.26$ | $83.54 \pm 0.24$ |
| | Input X Gradients | $82.98 \pm 0.37$ | $81.55 \pm 0.36$ | $77.23 \pm 0.40$ |
| Llama-2 | Integrated Gradients | $83.02 \pm 0.12$ | $80.42 \pm 0.17$ | $78.12 \pm 0.26$ |
| | LIME | $83.11 \pm 0.30$ | $82.54 \pm 0.061$ | $78.21 \pm 0.49$ |
| | Attention-SMaT | $92.81 \pm 0.53$ | $\underline{92.41} \pm 0.31$ | $88.10 \pm 0.44$ |
| | IvRA - Softmax | $\underline{92.76} \pm 0.21$ | $91.01 \pm 0.30$ | $\underline{88.83} \pm 0.31$ |
| | IvRA - Sparsemax | $\mathbf{94.31} \pm 0.35$ | $\mathbf{93.95} \pm 0.21$ | $\mathbf{89.43} \pm 0.49$ |

Table 8: Simulatability results for our experiments, expressed in accuracy %. **Bolded** values indicate the highest performance, with underlined values indicating the second highest performance.

|  |  | IMDb | SNLI | SQuAD |
|---|---|---|---|---|
| GPT-2 | Attention (Avg. all layers) | $0.109 \pm 0.118$ | $0.079 \pm 0.040$ | $-0.035 \pm 0.041$ |
|  | Attention (last layer) | $0.095 \pm 0.073$ | $0.035 \pm 0.036$ | $-0.022 \pm 0.044$ |
|  | Input X Gradients | $0.144 \pm 0.098$ | $0.135 \pm 0.019$ | $0.011 \pm 0.032$ |
|  | Integrated Gradients | $0.084 \pm 0.076$ | $0.144 \pm 0.061$ | $0.017 \pm 0.022$ |
|  | LIME | $0.085 \pm 0.035$ | $0.322 \pm 0.067$ | $0.084 \pm 0.053$ |
|  | Attention-SMaT | $0.244 \pm 0.027$ | $0.331 \pm 0.066$ | $\underline{0.123} \pm 0.046$ |
|  | IvRA - Softmax | $\mathbf{0.273} \pm 0.063$ | $\mathbf{0.386} \pm 0.068$ | $\mathbf{0.125} \pm 0.032$ |
|  | IvRA - Sparsemax | $\underline{0.266} \pm 0.029$ | $\underline{0.356} \pm 0.103$ | $0.119 \pm 0.052$ |
| Llama-2 | Attention (Avg. all layers) | $0.115 \pm 0.047$ | $0.099 \pm 0.066$ | $0.018 \pm 0.054$ |
|  | Attention (last layer) | $0.131 \pm 0.053$ | $0.104 \pm 0.075$ | $0.023 \pm 0.068$ |
|  | Input X Gradients | $0.149 \pm 0.059$ | $0.176 \pm 0.041$ | $0.094 \pm 0.104$ |
|  | Integrated Gradients | $0.141 \pm 0.034$ | $0.183 \pm 0.098$ | $0.086 \pm 0.042$ |
|  | LIME | $0.179 \pm 0.046$ | $0.355 \pm 0.037$ | $0.123 \pm 0.076$ |
|  | Attention-SMaT | $0.284 \pm 0.021$ | $0.364 \pm 0.069$ | $0.130 \pm 0.044$ |
|  | IvRA - Softmax | $\mathbf{0.325} \pm 0.068$ | $\mathbf{0.433} \pm 0.083$ | $\mathbf{0.151} \pm 0.080$ |
|  | IvRA - Sparsemax | $\underline{0.289} \pm 0.063$ | $\underline{0.362} \pm 0.028$ | $\underline{0.119} \pm 0.039$ |

Table 9: Comprehensiveness results for our experiments. **Bolded** values indicate the highest performance, with underlined values indicating second highest performance.

|  |  | IMDb | SNLI | SQuAD |
|---|---|---|---|---|
| GPT-2 | Attention (Avg. all layers) | $0.183 \pm 0.029$ | $0.635 \pm 0.015$ | $0.691 \pm 0.051$ |
|  | Attention (last layer) | $0.143 \pm 0.035$ | $0.747 \pm 0.035$ | $0.797 \pm 0.031$ |
|  | Input X Gradients | $0.219 \pm 0.033$ | $0.549 \pm 0.032$ | $0.775 \pm 0.044$ |
|  | Integrated Gradients | $0.197 \pm 0.022$ | $0.604 \pm 0.059$ | $0.622 \pm 0.04$ |
|  | LIME | $0.153 \pm 0.014$ | $0.442 \pm 0.036$ | $0.580 \pm 0.023$ |
|  | Attention-SMaT | $0.143 \pm 0.019$ | $\underline{0.409} \pm 0.068$ | $\underline{0.533} \pm 0.041$ |
|  | IvRA - Softmax | $\underline{0.136} \pm 0.041$ | $0.448 \pm 0.058$ | $0.565 \pm 0.015$ |
|  | IvRA - Sparsemax | $\mathbf{0.053} \pm 0.025$ | $\mathbf{0.347} \pm 0.04$ | $\mathbf{0.509} \pm 0.015$ |
| Llama-2 | Attention (Avg. all layers) | $0.180 \pm 0.008$ | $0.666 \pm 0.034$ | $0.763 \pm 0.013$ |
|  | Attention (last layer) | $0.111 \pm 0.018$ | $0.599 \pm 0.017$ | $0.799 \pm 0.013$ |
|  | Input X Gradients | $0.112 \pm 0.022$ | $0.489 \pm 0.042$ | $0.860 \pm 0.017$ |
|  | Integrated Gradients | $0.101 \pm 0.032$ | $0.467 \pm 0.04$ | $0.891 \pm 0.017$ |
|  | LIME | $\underline{0.099} \pm 0.06$ | $0.400 \pm 0.010$ | $0.645 \pm 0.004$ |
|  | Attention-SMaT | $0.113 \pm 0.027$ | $0.396 \pm 0.026$ | $0.656 \pm 0.016$ |
|  | IvRA - Softmax | $0.115 \pm 0.038$ | $0.386 \pm 0.044$ | $0.612 \pm 0.016$ |
|  | IvRA - Sparsemax | $\mathbf{0.016} \pm 0.013$ | $\mathbf{0.221} \pm 0.063$ | $\mathbf{0.423} \pm 0.153$ |

Table 10: Sufficieny results for our expreriments. For sufficiency, lower values indicate better performance. The best results are **bolded** and second-best results are underlined.

|  |  | IMDb | SNLI | SQuAD |
|---|---|---|---|---|
| GPT-2 | Attention (Avg. all layers) | $0.299 \pm 0.024$ | $0.145 \pm 0.03$ | $0.113 \pm 0.032$ |
|  | Attention (last layer) | $0.226 \pm 0.063$ | $0.111 \pm 0.016$ | $0.138 \pm 0.022$ |
|  | Input X Gradients | $0.268 \pm 0.108$ | $0.155 \pm 0.027$ | $0.149 \pm 0.018$ |
|  | Integrated Gradients | $0.259 \pm 0.142$ | $0.239 \pm 0.033$ | $0.146 \pm 0.098$ |
|  | LIME | $0.216 \pm 0.126$ | $0.206 \pm 0.010$ | $0.114 \pm 0.027$ |
|  | Attention-SMaT | $0.302 \pm 0.026$ | $0.236 \pm 0.010$ | $0.173 \pm 0.015$ |
|  | IvRA - Softmax | $\underline{0.322} \pm 0.041$ | $\mathbf{0.258} \pm 0.020$ | $\underline{0.176} \pm 0.052$ |
|  | IvRA - Sparsemax | $\mathbf{0.326} \pm 0.04$ | $\underline{0.240} \pm 0.006$ | $\mathbf{0.181} \pm 0.033$ |
| Llama-2 | Attention (Avg. all layers) | $0.372 \pm 0.021$ | $0.230 \pm 0.024$ | $0.185 \pm 0.014$ |
|  | Attention (last layer) | $0.365 \pm 0.019$ | $0.231 \pm 0.022$ | $0.194 \pm 0.019$ |
|  | Input X Gradients | $0.421 \pm 0.034$ | $0.321 \pm 0.012$ | $0.178 \pm 0.021$ |
|  | Integrated Gradients | $0.410 \pm 0.027$ | $0.327 \pm 0.017$ | $0.144 \pm 0.027$ |
|  | LIME | $0.385 \pm 0.017$ | $0.315 \pm 0.032$ | $0.178 \pm 0.028$ |
|  | SMaT | $0.422 \pm 0.031$ | $0.356 \pm 0.007$ | $0.287 \pm 0.027$ |
|  | IvRA - Softmax | $\underline{0.429} \pm 0.026$ | $\underline{0.357} \pm 0.015$ | $\mathbf{0.298} \pm 0.011$ |
|  | IvRA - Sparsemax | $\mathbf{0.430} \pm 0.008$ | $\mathbf{0.361} \pm 0.014$ | $\underline{0.289} \pm 0.022$ |

Table 11: Consistency results for our experiments. **Bolded** values indicate the highest performance, with underlined values indicating second highest performance.