# OpenReview forum: "IvRA: A Framework to Enhance Attention-Based Explanations for Language Models with Interpretability-Driven Training"
_EMNLP/2024/Workshop/BlackBoxNLP — BlackboxNLP 2024_

### Official Review · Reviewer_LcZt · 2024-09-02

**Overall Assessment:** 4
**Confidence:** 3

**Best Paper:**

2

**Best Paper Justification:**

The paper is extremely well-written, the tool it introduces can be useful for further research and the ablation studies provide some nice insights.

**Comments Questions Suggestions And Typos:**

l.019 "consistenti"

**Paper Summary:**

This paper adds an attention module, called IvRA, to an LM that is optimized with respect to four common interpretability measures: simulatability, comprehensiveness, sufficiency and consistency. The experiments cover three LMs and three text classification and answer span detection tasks and demonstrate that their IvRA method improves on all of them as compared to standard attention and other extractive explainability methods such as LIME and IG. Interestingly, the consistency optimization has a strong negative effect on the performance while the effect of sufficiency and simulatability is comparably small.

**Summary Of Strengths:**

This paper is thoughtful and well-written. It contains a comprehensive set of experiments.
The IvRA module can be practically useful for researchers who need explanations that satisfy one or more of the interpretability measures. The ablation studies provide good guidance for choosing the measures to optimize.

It's an insightful read for anyone interested in extractive explanations.

**Summary Of Weaknesses:**

One question that comes up to me relates to Goodhart's law: Do we consider these four measures that are used as the *definition* of interpretability, that are suitable as training objectives to increase interpretability, or are they rather *indicators*? This is a philosophical rather than a practical problem for your paper, but I'd like to see it discussed.

---

### Official Review · Reviewer_3vZC · 2024-09-08

**Overall Assessment:** 5
**Confidence:** 3

**Best Paper:**

2

**Best Paper Justification:**

Tackles many important desiderata of current NLP interpretability research. Extensive and exciting results.

**Comments Questions Suggestions And Typos:**

* l. 019: "consistent~~i~~"
* Figure 2 should explain what the variables represent. There are many labels missing that are hard to match between full text and figure.
* Figure 4 caption: "highlightin**g**"
* l. 470: "three ~~three~~"
* Depiction of $t^k_i$ in l.244: Format like Eq. 8 etc.
* Apply same highlighting of criteria to §2.2 as in Table 5

**Paper Summary:**

The paper is about exploring whether attention-based explanations can be adapted to meet the demands of interpretability paradigms (simulatability, faithfulness, consistency). The authors present IvRA which regularizes a language model's attention distribution s.t. attributions are guided to align more with these desiderata.

**Summary Of Strengths:**

* Addresses many concerns in interpretability research and the methodology is well-founded and in agreement with recent research.
* Well-written, easy-to-understand explanations, well-documented. Images and graphics aid the reader's understanding.
* The extensive results show how different XAI paradigms are at odds with each other and the authors give a reasonable explanation why that is the case.
* It shows that using criterion-aligned attention scores is useful and competitive with typical post-hoc methods as strong baselines and surpasses them in many regards such as efficiency.
* Ablation studies such as mixed-criteria training and impact of IvRA on downstream task accuracy provide further insights and make the study more rigorous and trustworthy.

**Summary Of Weaknesses:**

Almost none, but I'd suggest that the authors look into polishing the figures and explanations of the mathematical formalization (see Comments).

---

### Decision · Program_Chairs · 2024-09-17

**Decision:**

Accept

**Comment:**

Reviewers agree that this paper is impactful, relevant, and well-written. There were minor suggestions regarding presentation and framing, but this is already a strong contribution that fits the theme of the workshop well.